# Activity dynamics of amygdala GABAergic neurons during cataplexy of narcolepsy

**Ying Sun, Carlos Blanco-Centurion, Emmaline Bendell, Aurelio Vidal-Ortiz, Siwei  Luo, Meng Liu***

Department of Psychiatry and Behavioral Sciences, Medical University of South Carolina, Charleston, United States

**Abstract** Recent studies showed activation of the GABAergic neurons in the central nucleus of the amygdala (CeA) triggered cataplexy of sleep disorder narcolepsy. However, there is still no direct evidence on CeA GABAergic neurons' real-time dynamic during cataplexy. We used a deep brain calcium imaging tool to image the intrinsic calcium transient as a marker of neuronal activity changes in the narcoleptic VGAT-Cre mice by expressing the calcium sensor GCaMP6 into genetically defined CeA GABAergic neurons. Two distinct GABAergic neuronal groups involved in cataplexy were identified: spontaneous cataplexy-ON and predator odor-induced cataplexy-ON neurons. Majority in the latter group were inactive during regular sleep/wake cycles but were specifically activated by predator odor and continued their intense activities into succeeding cataplexy bouts. Furthermore, we found that CeA GABAergic neurons became highly synchronized during predator odor-induced cataplexy. We suggest that the abnormal activation and synchronization of CeA GABAergic neurons may trigger emotion-induced cataplexy.
DOI: https://doi.org/10.7554/eLife.48311.001

## Introduction

Narcolepsy is a chronic sleep disorder characterized by excessive daytime sleepiness, cataplexy, sleep fragmentation, and hypnogogic/hypnopompic hallucinations. Cataplexy is cardinal among these symptoms and characterized by a sudden loss of skeletal muscle tone during waking.

Even though it is known that the loss of the neuropeptide orexin (hypocretin, HCRT) system causes narcolepsy with cataplexy (*Lin et al., 1999*; *Nishino et al., 2000a*), the entire brain circuitry responsible for the presentation of all narcoleptic symptoms is not fully understood. An important clue to unraveling the brain circuit of narcolepsy comes from the fact that cataplexy is usually triggered by strong emotions (*Dauvilliers et al., 2014*; *Morawska et al., 2011*). In fact, there is some evidence pointing to the involvement of the amygdala as part of the narcolepsy circuit, whereas the exact circuitry involved and the specific abnormalities in the amygdala are still unknown. A single unit recording study done in narcoleptic dogs found that, during cataplexy, some amygdala neurons become activated (*Gulyani et al., 2002*). However, the intrinsic technical limitations of the single-unit recordings prevented the identification of the phenotypes of those neurons activated during cataplexy. To overcome this limitation, in the present study, we took advantage of molecular genetic tools to both tag and measure the in vivo activity of CeA GABAergic neurons from transgenic narcoleptic mice. We first crossed vesicular GABA transporter *Cre* mice (*Slc32a1-ires-Cre*, or VGAT-Cre mice) with orexin KO mice (*Hcrt $^{-/-}$*), to generate VGAT-Cre narcoleptic mice (*Slc32a1-ires-Cre$^{+/-}$/Hcrt$^{-/-}$*). Later, we transfected CeA GABAergic neurons with genetically controlled calcium ($Ca^{2+}$) sensor GCaMP6, which could be imaged via a GRIN lens embedded in the amygdala and a miniature microscopic camera. Predator odor coyote urine was used to trigger emotion-induced cataplexy. We monitored and calculated the intracellular $Ca^{2+}$ transients because they are reliable readouts of the excitability level of neurons (*Chen et al., 2018*; *Chen et al., 2013*). We wanted to know what, in the

**\*For correspondence:**
liumen@musc.edu

**Competing interests:** The authors declare that no competing interests exist.

absence of orexin modulation, could be the abnormality in the CeA GABAergic neurons and how it is associated with the timing of cataplexy.

## Results

### Orexin immunostaining

All mice from the control group (*Slc32a1-ires-Cre$^{+/-}$/Hcrt $^{+/-}$*) showed abundant orexin immunoreactive neurons. In contrast, orexin immunoreactive neurons were completely absent in the group of narcoleptic mice (*Slc32a1-ires-Cre$^{+/-}$/Hcrt $^{-/-}$*) (*Figure 1, A and B*). These results, together with the genotyping results and signature cataplexy behaviors observed, validated the animal model of narcolepsy used in this study.

### Anatomical distribution of GCaMP6 expression

Only mice showing correct vector and GRIN lens targeting were chosen for further data analysis. This on-target group was made up of 5 narcoleptic mice and five control mice. In these mice, GCaMP6s predominantly expressed within the CeA area (>80%), but some scattered expression was also observed within the basolateral amygdala (BLA) and the basomedial amygdala (BMA) (*Figure 1D*). VGAT immunostaining results showed that around 95% GCaMP6 expressing neurons in CeA also contained VGAT in the cytoplasm (*Figure 1, E-G*, and *Figure 1—source data 1*).

### Amygdala GABAergic neuronal activity during undisturbed recording and predator odor exposure

Altogether, 186 GABAergic cells from five narcoleptic mice and 207 GABAergic cells from five control mice were imaged (*Figure 2*, *Video 1*, and *Table 1* for numbers of recorded cells from each mouse). GLMM analysis failed to detect a significant fixed (group) effect, that is narcoleptic vs. control ($F_{(1, 2325)}$=0.0010, p=0.971). Yet it found a significant effect on the calcium fluorescent intensity ∆F/F Z-scores ($F_{(6, 2325)}$=318.02, p<0.001) depending on individual sleep/wake states. In both groups of mice, significantly higher average Z-scores were observed during active waking (AW) and rapid eye movement sleep (REMS) as compared to either quiet waking (QW) or non-rapid eye movement sleep (NREMS) (*Figure 2C*). As for the control group of mice, 24.64% (51/207) were REMS-ON, 29.47% (61/207) were AW-ON, 4.83% (10/207) were REMS/AW-ON, and 3.38% (7/207) were NREMS-ON. Percentages of ON cells in each sleep state were similar among narcoleptic mice: 23.66% (44/186) were REMS-ON, 31.18% (58/186) were AW-ON, 6.99% (13/186) were REMS/AW-ON, and 3.76% (7/186) were NREMS-ON. Additionally, in the narcoleptic group, 8.06% (15/186) were identified as spontaneous cataplexy-ON (SC-ON) cells, with 11 of these cells exclusively activated during cataplexy, while the other four also activated during NREMS (*Figure 2D*, showing one SC/NREMS-ON cell). We also found that 47.34% (98/207) of cells in the control group and 42.47% (79/186) of cells in the narcoleptic group kept relatively moderate fluctuations of Ca$^{2+}$ signals and, consequently, could not be scored as 'ON' cells in any brain states during the recording before predator odor exposure. We named these cells 'unscored cells' accordingly (*Table 2* and *Figure 2D*). However, many of these unscored cells presented dramatic activity changes in response to predator odor and during predator odor-triggered cataplexy in the narcoleptic group (*Table 2*, *Figure 3*, C and F).

Upon the first 3 min of exposure to the predator odor, both the control and the narcoleptic mice showed many typical behavioral signs of novelty and fear including direct examination of the odor source, avoidance and escaping attempts (*Video 2*). As for the control mice, the average Z-scores were slightly elevated compared to the undisturbed AW episode, but no significance was detected (Z-score: 0.44 ± 0.028 before odor versus 0.48 ± 0.031 after odor, $t_{(4, 2325)}$=1.20, p=0.46, *Figure 2C*). ON cell percentage was 29.47% (61/207) before odor (undisturbed AW) and 31.88% (66/207) after odor (AW after odor exposure, AWO). In contrast, among the narcoleptic mice, the average Z-scores were significantly higher during the first 3 min of odor exposure, increasing from 0.41 ± 0.027 to 0.66 ± 0.029 ($t_{(6, 2325)}$=7.21, p<0.001, *Figures 2C* and *4G*). The percentage of ON cells almost doubled: 31.38% (58/186) before odor versus 60.22% (112/186) after odor. In the narcoleptic group, if an AWO's Z-score was statistically higher than the average Z-score of all undisturbed AW bouts (one-sample T-test, p<0.05), this AWO was defined as a hyperactive AW (HAW). We then

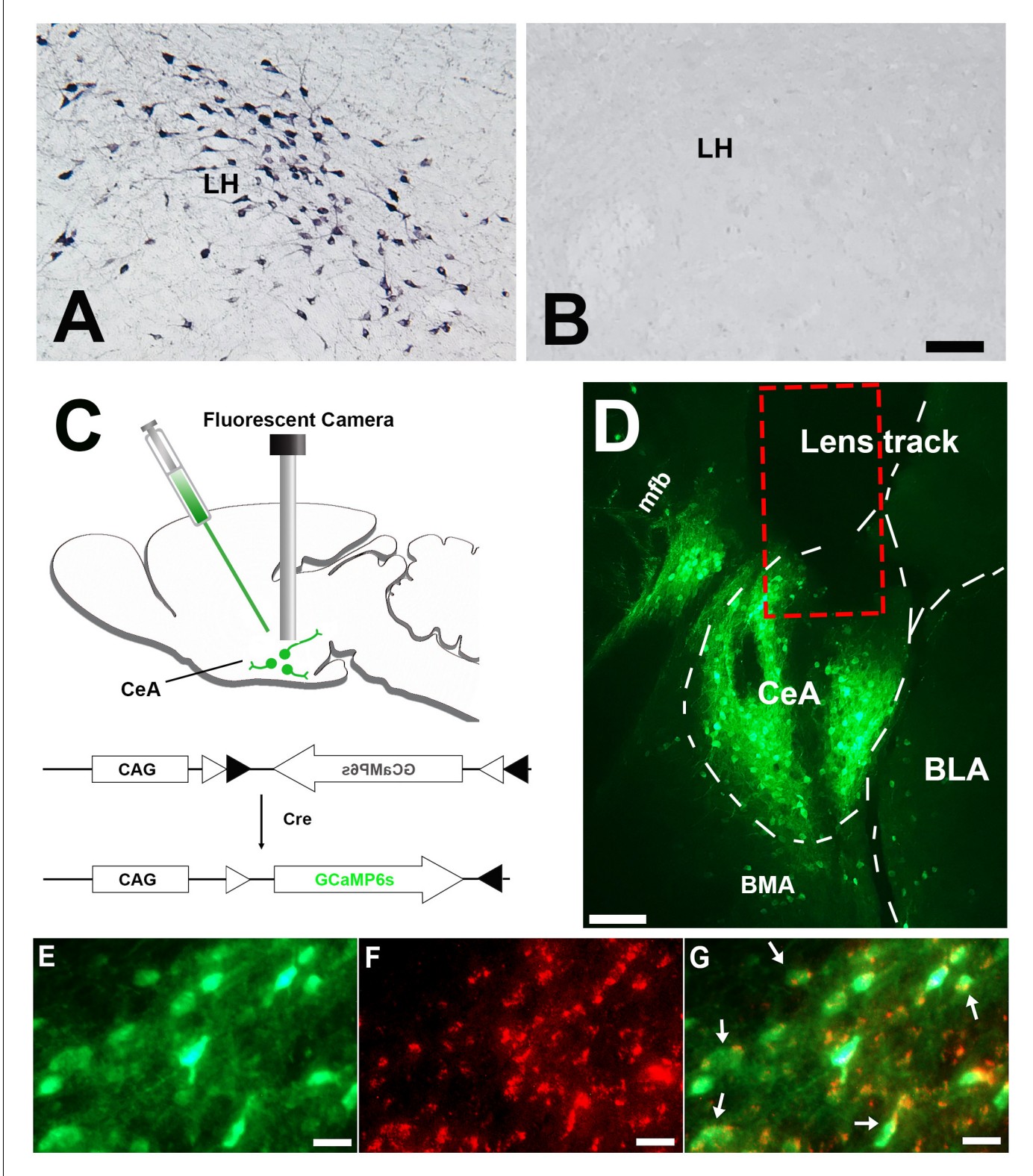

**Figure 1.** Histology results. (**A**) Immunostaining demonstrating abundant orexin immunoreactive neurons and fibers in the LH of the control mice (n = 8). (**B**) orexin immunoreactive somata and fibers were utterly absent in LH of the narcoleptic VGAT-Cre mice, n = 10). (**C**) Illustration of vector injection and miniature camera installation. (**D**) Lens track and abundant GCaMP6 expression in CeA and adjacent areas. (**E**) GCaMP6 expression in neuronal somata and axons in CeA. (**F**) cytoplasm VGAT immunoreactivities in the same area as E. (**G**) Co-localization of GCaMP6 and VGAT. About 95% GCaMP6 expressing neurons in CeA also contain VGAT immunoreactivities (arrows point to some of the double-labeled neurons). CeA: the central

*Figure 1 continued on next page*

*Figure 1 continued*

nucleus of the amygdala. BLA: basolateral amygdala. BMA: basomedial amygdala. LH: lateral hypothalamus. mfb: medial forebrain bundle. Scale bars in A, D = 50 µm. Scale bar in E-G = 10 µm.

DOI: https://doi.org/10.7554/eLife.48311.002

The following source data is available for figure 1:

**Source data 1.** Cell counts of neurons expressing GCaMP6 and VGAT in the amygdala of one set of coronal sections from the narcoleptic mice used for calcium data analysis (n = 5).

DOI: https://doi.org/10.7554/eLife.48311.003

defined cataplexy bouts preceded by such HAW episode as emotion-induced cataplexy (EC). Otherwise, they were defined as spontaneous cataplexy (SC). In other words, EC bouts always came after HAW episodes, whereas SC bouts were always preceded by an undisturbed AW episode. We observed that many HAW-ON cells reached their maximal fluorescent intensity level upon odor exposure and during the following EC bouts (*Figure 4, D and F*). In the total of 112 HAW-ON cells, 44.64% (50/112) were from those 'unscored cells,' and 35.71% (40/112) were from AW-ON cells. The remainder consisted of portions of cells activated in other states, including REMS, NREMS, and SC (*Table 2*).

## Amygdala GABAergic neuronal activity during cataplexy

Altogether, we recorded 10 bouts of SC and 14 bouts of EC. Neither SC nor EC was observed in mice of the control group. $Ca^{2+}$ signal imaged during SC was significantly lower than that measured during REMS and AW while $Ca^{2+}$ signal during EC was at the level of undisturbed AW and markedly higher than during SC (*Figure 2C*). Out of the 186 recorded cells, 15 (8.06% of 186) were scored as SC-ON and 74 (39.78% of 186) were scored as EC-ON (*Figure 3,A-C*). Based on neuronal activity during other sleep states, we found that SC-ON and EC-ON neurons are a distinct group of neurons, and there was no overlap in between. Thus, among the 74 EC-ON neurons, 54.05% (40/74) came from the 'unscored cells,' 22.97% (17/74) from AW-ON cells and 22.97% (17/74) were made up of REMS-ON or REMS/AW-ON cells (*Figure 3*, *Table 2*). 'Unscored cells' presented relatively higher $Ca^{2+}$ activity during the HAW-EC transition episodes and the following cataplexy bouts, compared to REMS-ON and AW-ON cells (*Figure 3, D-F*). *Figure 4* demonstrated that many EC-ON neurons reached their maximal activity level after coyote urine exposure and continued the maximal activity level into following cataplexy bouts (*Video 3 and Video 4*: animal behavior during SC and EC bouts in *Figure 4, C and D*). The actual calcium intensity ΔF/F Z-score changes were plotted in the *Figure 4—figure supplement 1*, to show the dramatic increase of Z-score upon odor exposure in the narcoleptic mice.

To further characterize the hyperactivity of HAW episodes in the narcoleptic mice, we calculated the amplitude and frequency of the prominent calcium transient peaks and found both parameters were significantly increased after coyote urine exposure. The cumulative probability test indicated that the peak frequency was dramatically shifted to a higher rate (*Figure 4, H-J*).

## Amygdala GABAergic network activity patterns during cataplexy

To explore the GABAergic network activity patterns during sleep/wake states and cataplexy, we examined the neuronal activity correlation (represented by the corrected Pearson correlation coefficient, $Z_R$), then created a spatial connectivity map by combining the cell spatial references and $Ca^{2+}$ fluorescent intensity (Z-scored ΔF/F) correlations among cells. In the control group, the average positive $Z_R$ values of recorded cells kept at a relatively stable level among various sleep/wake states (0.10 ± 0.0068 at AW, 0.087 ± 0.0054 at QW, 0.12 ± 0.017 at NREMS, 0.096 ± 0.012 at REMS). Predator odor exposure produced only an insignificant increase in $Z_R$ (AWO: 0.13 ± 0.017). In contrast, $Z_R$ values in the narcoleptic mice were much lower at normal sleep/wake states (0.032 ± 0.0037 at AW, 0.028 ± 0.0028 at QW, 0.038 ± 0.0047 at NREMS, 0.035 ± 0.0033 at REMS, 0.027 ± 0.0019 at HAW). Two-way ANOVA analysis found a significant difference in $Z_R$ between the control and narcoleptic groups ($F_{(1,64)}$=57.28, p=0.004).

During SC, $Z_R$ was brought up to 0.14 ± 0.014, which was already significantly higher than any other sleep/wake states (p<0.001) in the narcoleptic mice. Upon exposure to predator odor, $Z_R$

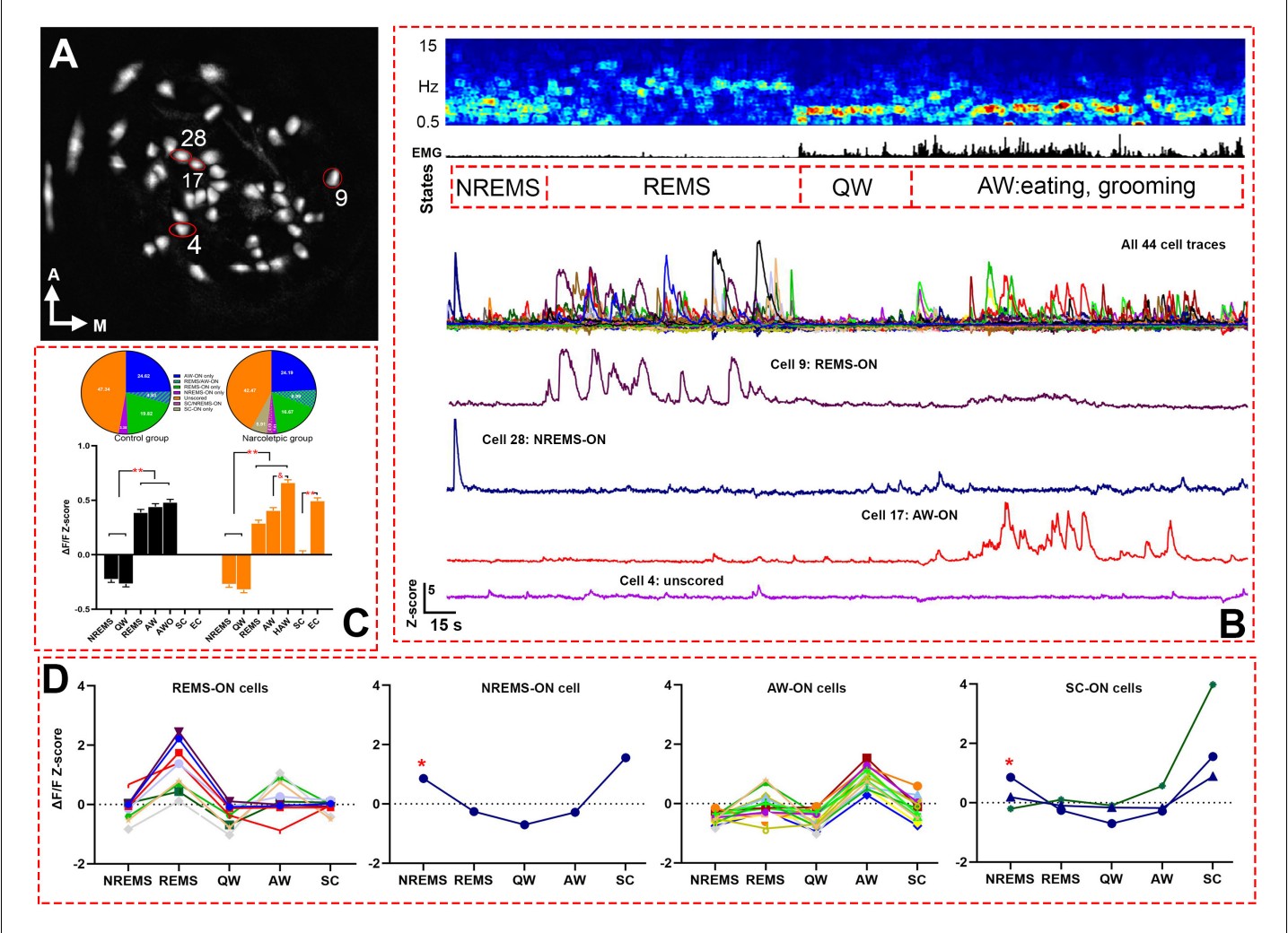

**Figure 2.** Live Ca²⁺ activity recording and cell classification. (**A**) Map of 44 recorded cells from narcoleptic mouse LR02082418. (**B**) A serial figures showing (top to bottom) EEG spectrogram, EMG, brain states, cell traces of a total of 44 cells, cell traces of four example cells outlined in A. (**C**) bar graph (bottom) of Z-scores in each brain state and Bonferroni pairwise comparison results, and pie charts (top) of ON cell percentages in each brain states. **: $p < 0.001$ compared to NREMS and QW in both groups, or SC in the narcoleptic group. and : $p < 0.001$ compared to AW in the narcoleptic group. (**D**) Z-scores graphs of 'ON' cells in each brain state during undisturbed recording period in mouse LR02082418, including 10 REMS-ON, 1 NREMS-ON, 15 AW-ON, and 3 SC-ON cells. The only NREMS-ON cell (marked with *) was also one of the SC-ON neurons. AWO: active waking after odor exposure in control mice. HAW: hyperactive AW after odor exposure in the narcoleptic mice. SC: spontaneous cataplexy during undisturbed recording. EC: emotion-induced cataplexy during odor exposure.

DOI: https://doi.org/10.7554/eLife.48311.004

The following source data is available for figure 2:

**Source data 1.** Z-score data for *Figure 2, C and D*.
DOI: https://doi.org/10.7554/eLife.48311.005

stayed low during HAW ($0.027 \pm 0.0019$) until EC bouts occurred. $Z_R$ during EC was elevated abruptly to $0.37 \pm 0.046$ ($p < 0.001$ compared to other states of both groups) (*Figure 5*).

## Discussion

Ours is the first study to image in vivo Ca²⁺ transients in individual CeA GABAergic neurons during sleep/wake states, spontaneous cataplexy and emotion(fear)-induced cataplexy. We found abnormal hyperactivity during active waking episodes and hyper synchronization during succeeding emotion-

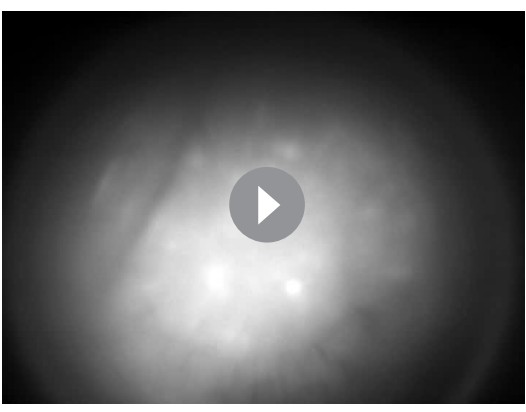

**Video 1.** A sample of raw calcium movies played at 64x speed.
DOI: https://doi.org/10.7554/eLife.48311.010

induced cataplexy from amygdala GABAergic neurons. The following are the five important implications from this study.

## CeA GABAergic neurons are more active during REMS and AW

Past studies found the amygdala can modulate sleep states. Early in 1975, researchers (*Smith and Miskiman, 1975*) found that electrical stimulation of the amygdala significantly increased REMS. Likewise, electrophysiological recordings showed that amygdala neurons become very active during waking and REMS, whereas displayed the lowest activity during NREMS (*Corsi-Cabrera et al., 2016*; *Muñoz-Torres et al., 2018*). Our current study using in vivo imaging of individual $Ca^{2+}$ transients in GABAergic neurons corroborates those electro-physiological recordings.

We found that in both control and narcoleptic mice, CeA GABAergic neurons showed their maximal activities during AW and REMS. During these two states, these neurons displayed significantly higher $Ca^{2+}$ transients and larger percentages of active neurons as compared to QW and NREMS (*Figure 2*). We also observed that neurons active during REMS are a distinct group of cells from those neurons that become active during AW. Less than 7% of all imaged neurons were equally active during both REMS and AW. The segregation in the activity profile during either state suggests there might be two distinct neural pathways activating specific groups of CeA GABAergic neurons. For instance, we observed that GABAergic AW-ON neurons became maximally active during eating. This observation is consistent with the hypothesis claiming that amygdala GABAergic neurons regulate food consumption through a positive-valence circuit (*Douglass et al., 2017*). In contrast, REMS-ON GABAergic neurons might be part of the amygdala-related memory processing circuit (*Genzel et al., 2015*).

We found that the average $Ca^{2+}$ signal and the percentages of active neurons in each sleep/wake state were similar between the control and narcoleptic mice. The normal activity of CeA GABAergic neurons during sleep/wake states measured in the narcoleptic mice suggests these neurons function normally outside of conditions leading to emotionally-triggered cataplexy.

On the other hand, we also found, in both groups, a particular group of CeA GABAergic neurons (unscored cells) with relatively low activity and insignificant activity changes during undisturbed sleep/wake states. In other words, it is unnecessary for all of the GABAergic neurons to be activated

**Table 1.** Number of recorded cells in each mouse.

| Mouse ID | Sex | Group | Numbers of cells |
|---|---|---|---|
| LR00062218 | Male | Narcoleptic | 27 |
| LR01062918 | Female | Narcoleptic | 35 |
| LR02082418 | Male | Narcoleptic | 44 |
| LR03091618 | Male | Narcoleptic | 42 |
| LR08012219 | Female | Narcoleptic | 38 |
| LR04101618 | Male | Control | 38 |
| LR12011419 | Male | Control | 28 |
| LR13022419 | Female | Control | 63 |
| LR14022819 | Female | Control | 35 |
| LR16031319 | Male | Control | 43 |

DOI: https://doi.org/10.7554/eLife.48311.008

**Table 2.** components of HAW-ON and EC-ON neurons in the narcoleptic group.

| | Unscored | AW-ON | REMS-ON REMS/AW-ON | NREMS-ON SC-ON | EC-ON | HAW-ON |
|---|---|---|---|---|---|---|
| HAW-ON (112) | 50 (44.64%) | 40 (35.71%) | 13 (11.61%) | 8 (7.14%) | 63 (56.25%) | ——— |
| EC-ON (74) | 40 (54.05%) | 17 (22.97%) | 17 (22.97%) | 0 | ——— | 63 (85.14%) |

DOI: https://doi.org/10.7554/eLife.48311.009

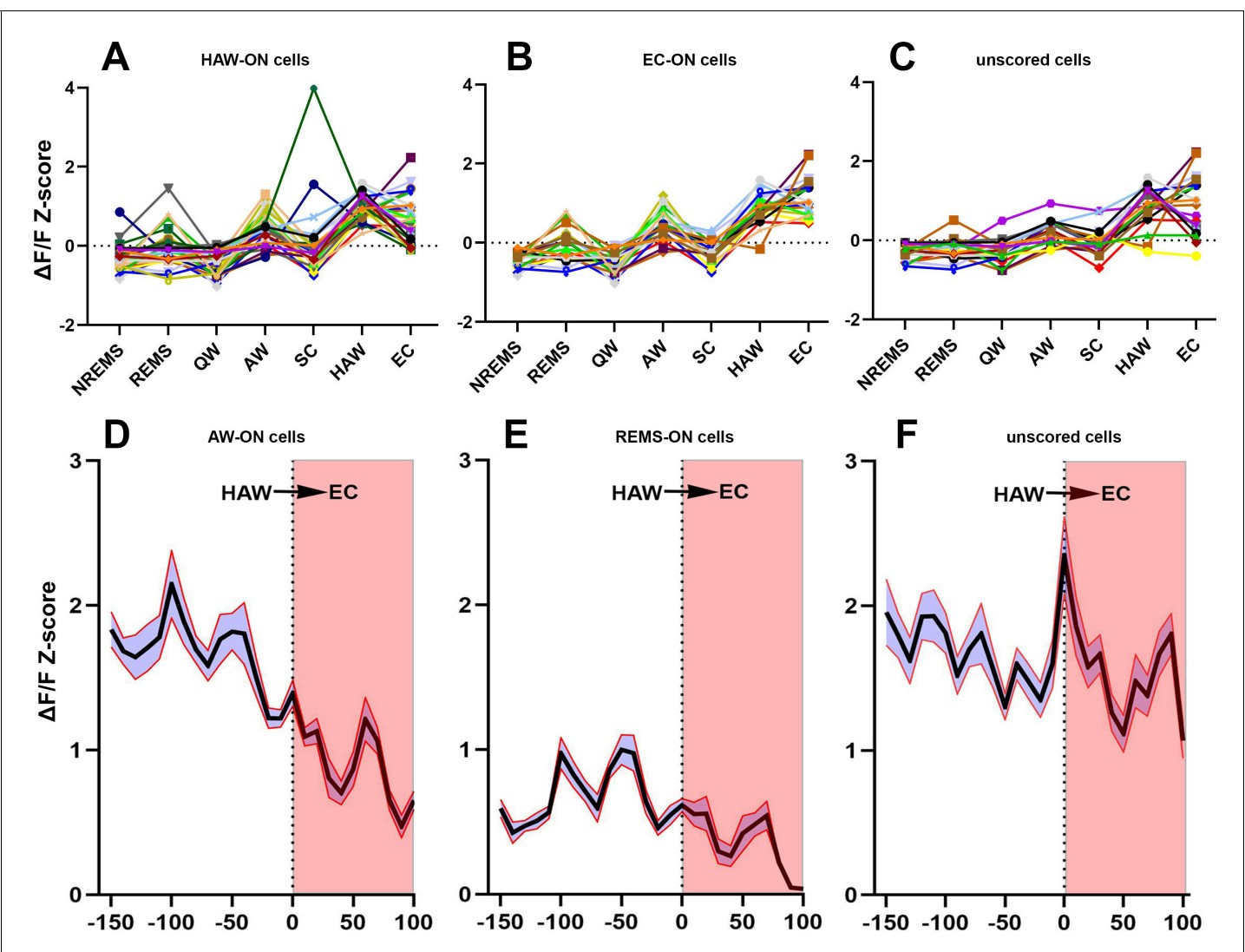

**Figure 3.** Activities of sorted neurons during sleep/wake cycle and cataplexy. (A–C) Neuronal activity graphs of 28 HAW-ON cells, 19 EC-ON cells and 17 unscored cells in the narcoleptic mouse LR02082418. Most of the unscored cells displayed significantly elevated activities during HAW and EC (C). (D–F) average Z-score changes during HAW–EC transition from 112 HAW-ON cells, 44 REMS-ON cells, and 79 unscored cells in the narcoleptic group. The durations of cataplexy bouts were normalized as a percentage between 0–100. The average activity of HAW-ON cells stayed high upon odor exposure but gradually decreased during the transition into EC (D). Compared to the low activity of REMS-ON cells (E), unscored cells displayed stronger activation upon odor exposure and maintained the high activity level during EC (F).

DOI: https://doi.org/10.7554/eLife.48311.006

The following source data is available for figure 3:

**Source data 1.** Z-score data for transition map.

DOI: https://doi.org/10.7554/eLife.48311.007

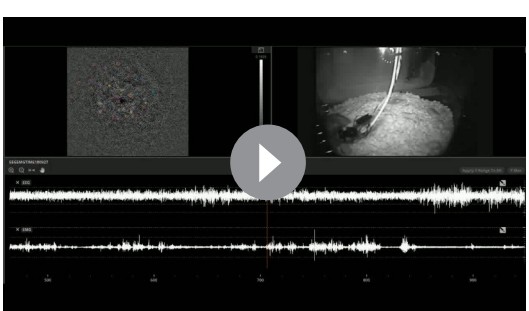

**Video 2.** Behaviors upon coyote urine exposure from a control mouse (Left) and a narcoleptic mouse (right). DOI: https://doi.org/10.7554/eLife.48311.014

during normal sleep/wake states, suggesting there might be a modulating mechanism preventing the amygdala from being overactive. Indeed, this modulating mechanism might be even stronger in wild type mice because exposure to predator odor only awakened a small portion of these 'unscored cells'. In contrast, among the narcoleptic group, the majority of these 'unscored cells' became highly activated upon odor exposure and actively participated in the subsequent emotion-induced cataplexy. The absence of orexin might be the main reason for the failure of this modulation.

## Narcoleptic mice CeA GABAergic neurons become hyperactive during predator odor exposure

Although many studies have demonstrated the importance of the amygdala on narcolepsy and cataplexy (*Burgess et al., 2013*; *Liu et al., 2016*; *Mahoney et al., 2017*; *Snow et al., 2017*), the precise causal role of the amygdala in narcolepsy is still unknown. Functional imaging studies on human narcolepsy have given inconsistent results. Some studies observed reduced activity and volume within the amygdala while others found increased activity and unchanged volume (*Brabec et al., 2011*; *Ponz et al., 2010*; *Vaudano et al., 2019*; *Wada et al., 2019*). The discrepancy may arise from either the individual disease seriousness among different patients or the limitations of current functional imaging tools used clinically, which do not have the single-cell resolution or cannot measure directly neuronal activity. Our study has overcome both technical barriers, and because we used a genetic animal model of narcolepsy, we also had a consistent level of symptomatology. We are providing the very first direct evidence that CeA GABAergic neurons of narcoleptic mice become hyperactive upon presentation of an innate fearful stimulus. Control mice did not become hyperactive, whereas both narcoleptic and control mice showed avoidance and fear behaviors. CeA GABAergic hyperactivity found in the narcoleptic mice had both increased recruitment (percentage of AW-ON neuron doubled) and stronger facilitation (higher $Ca^{2+}$ Z-scores, higher peak amplitude and frequency). In other words, aside from increasing activity intensity in regular AW-ON neurons, many additional neurons that were otherwise 'unscored' or active during REMS, became active in response to predator odor. 'Unscored cells' themselves made up 44.64% (50/112) of all activated neurons upon odor exposure (*Table 2*).

In contrast, when orexin modulation is undisturbed, neither the percentage of AW-ON neurons nor their activity levels change significantly after exposure to the same predator odor. Activities of most 'unscored cells' did not change significantly. Because CeA GABAergic neurons did not display hyperactivity unless narcoleptic mice had exposure to predator odor, it strongly indicates that orexin plays a vital modulating role during strong emotional states. When orexin is missing, the functional compensation effects from other neuropathways might prevent amygdala GABAergic neurons from behaving over actively until emotion stimuli come in.

Prior evidence indicated that orexin inhibits amygdala neurons by exciting serotoninergic terminals within the CeA to block cataplexy (*Hasegawa et al., 2017*). In this way, serotonin keeps GABAergic neuronal activity in check. Orexin can also excite inhibitory interneurons in the CeA (*Dustrude et al., 2018*; *Hasegawa et al., 2017*). GABAergic hyperactivity once triggered, could successively send overwhelming inhibitory inputs into the brainstem and cortex. Particularly, when GABAergic inhibition occurs in motoneurons, muscle weakness, or

**Video 3.** Demonstrating the spontaneous cataplexy (SC) bout in *Figure 4C*, 8x speed. DOI: https://doi.org/10.7554/eLife.48311.015

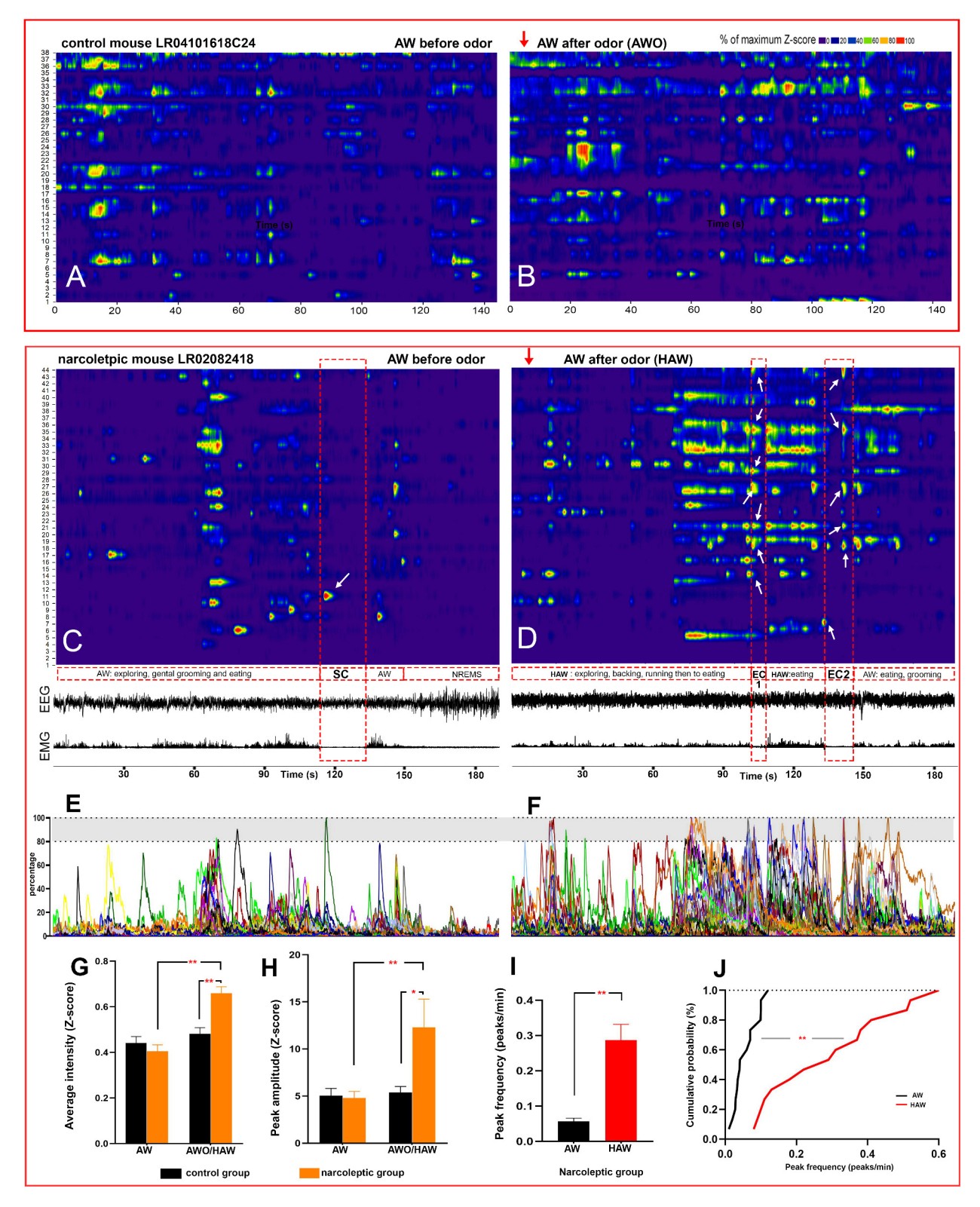

**Figure 4.** Hyperactivity of amygdala GABAergic neurons before and during emotion-induced cataplexy. (A–D), heat maps of $Ca^{2+}$ fluorescent intensity changes before and after coyote urine exposure (red arrows show the starting time of exposure) in control mouse LR04101618. (**A and B**) and narcoleptic mouse LR02082418 (**C, D**). The control mouse displayed moderate increases on activity (**B**) upon exposure to coyote urine. In contrast, the narcoleptic mouse showed dramatic activity changes, with significantly more neurons reaching their maximal activities during the first 3 min of exposure

*Figure 4 continued on next page*

*Figure 4 continued*

(**D and F**). A spontaneous cataplexy bout (SC) following a regular AW episode and an SC-ON cell (arrow in C) are shown in (**C**). Two emotion-induced cataplexy episodes (EC1 and EC2) closely following hyperactive AW (HAW) bouts induced by coyote urine are shown in (**D**), in which many EC-ON cells reached their maximal activities (arrows). (**E, F**): Graphs of the percentage to its maximal activity Z-score of each cell in (**C**) and (**D**), respectively. Prominent calcium transient peaks (examples are shown in the shadow areas of E and F) were selected for statistical analysis. Odor exposure significantly increased the average calcium signal intensity (**G**), peak amplitude (**H**), and peak frequency (**I**) in the narcoleptic group (*: p<0.05; **: p<0.001). (**J**): Cumulative probability function shows a significant shift toward a much higher peak frequency after coyote urine exposure in the narcoleptic mice (Matlab Kolmogorov-Smirnov Test: k = 0.73, p<0.001).

DOI: https://doi.org/10.7554/eLife.48311.011

The following source data and figure supplement are available for figure 4:

**Source data 1.** Percentage source data for *Figure 4* heat maps and graphs (**A–D, I, J**).

DOI: https://doi.org/10.7554/eLife.48311.013

**Figure supplement 1.** Calcium transient intensity ΔFF (Z-score) graphs (1-4) are incorporated into each heat maps in *Figure 4, A-D*, respectively, to demonstrate the dramatic intensity increase in the narcoleptic mice after coyote urine exposure.

DOI: https://doi.org/10.7554/eLife.48311.012

even paralysis may arise. Interestingly, our group recently identified a group of GABAergic bifurcating neurons in CeA and the basolateral amygdala (BLA) that simultaneously project to the medial prefrontal cortex (mPFC) and the vlPAG (*Sun et al., 2019*). Because vlPAG is a critical hub for controlling muscle tone (*Weber et al., 2018*), excessive GABAergic input to the vlPAG may lead to reduced motor output. Likewise, excessive GABAergic tone may be the cause of the hypo-excitability of motor cortical areas observed in narcolepsy (*Nardone et al., 2010*; *Weber et al., 2018*).

## Cataplexy-ON GABAergic neurons in the CeA of narcoleptic mice

CeA GABAergic neurons that become active during both spontaneous (SC) and emotion-induced cataplexy (EC) were identified. The SC-ON neurons either were exclusively activated in SC or also activated in NREMS, indicating that SC may share part of the neural substrate with NREMS, but not REMS, AW or EC, because none of the SC-ON neurons became active in above three states. However, EC-ON neurons had a wide range of sources, with about 50% from unscored cells, 23% from AW-ON cells and 23% from REMS-ON cells. Many EC-ON neurons were maximally activated during the preceding hyperactive AW (HAW) episode and continued showing intense activity until the end of the cataplexy bout (*Figure 4*, D and F, *Figure 4—figure supplement 1*). The transition activity analysis indicated that it was the 'unscored cells' that contributed most to the high activity level during EC. Thus, it is intriguing to track down the potential targets of these 'unscored' EC-ON cells and to see how it may trigger cataplexy.

Usually, cataplexy episodes that happen without any exogenous emotion stimulus are defined as SC, while those induced by emotion stimuli such as palatable food or aversive odor are defined as EC. However, neither behavioral nor neural mechanism criteria have been available for distinguishing SC and EC. We, for the first time, observed the potential neural substrate differences in between at the amygdala level, by using the HAW episode as the marker of EC.

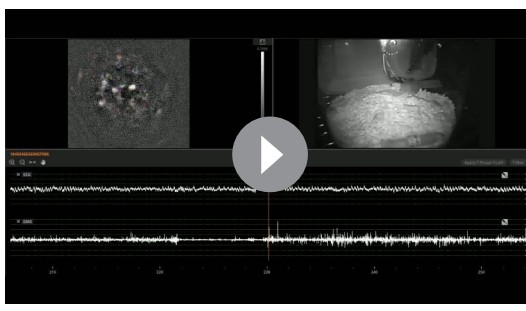

**Video 4.** Demonstrating the emotion-induced cataplexy (EC1) bout in *Figure 4D*, 8x speed.

DOI: https://doi.org/10.7554/eLife.48311.016

## Hyper synchronization of CeA GABAergic neurons during cataplexy

In addition to activity intensity, the coordination patterns among neurons are crucial for brain functions. The stable, intermediate cross-correlations observed among CeA GABA neurons in the control mice might be necessary for maintaining the amygdala's normal functions. Instead of being a pure stimulating or inhibiting source,

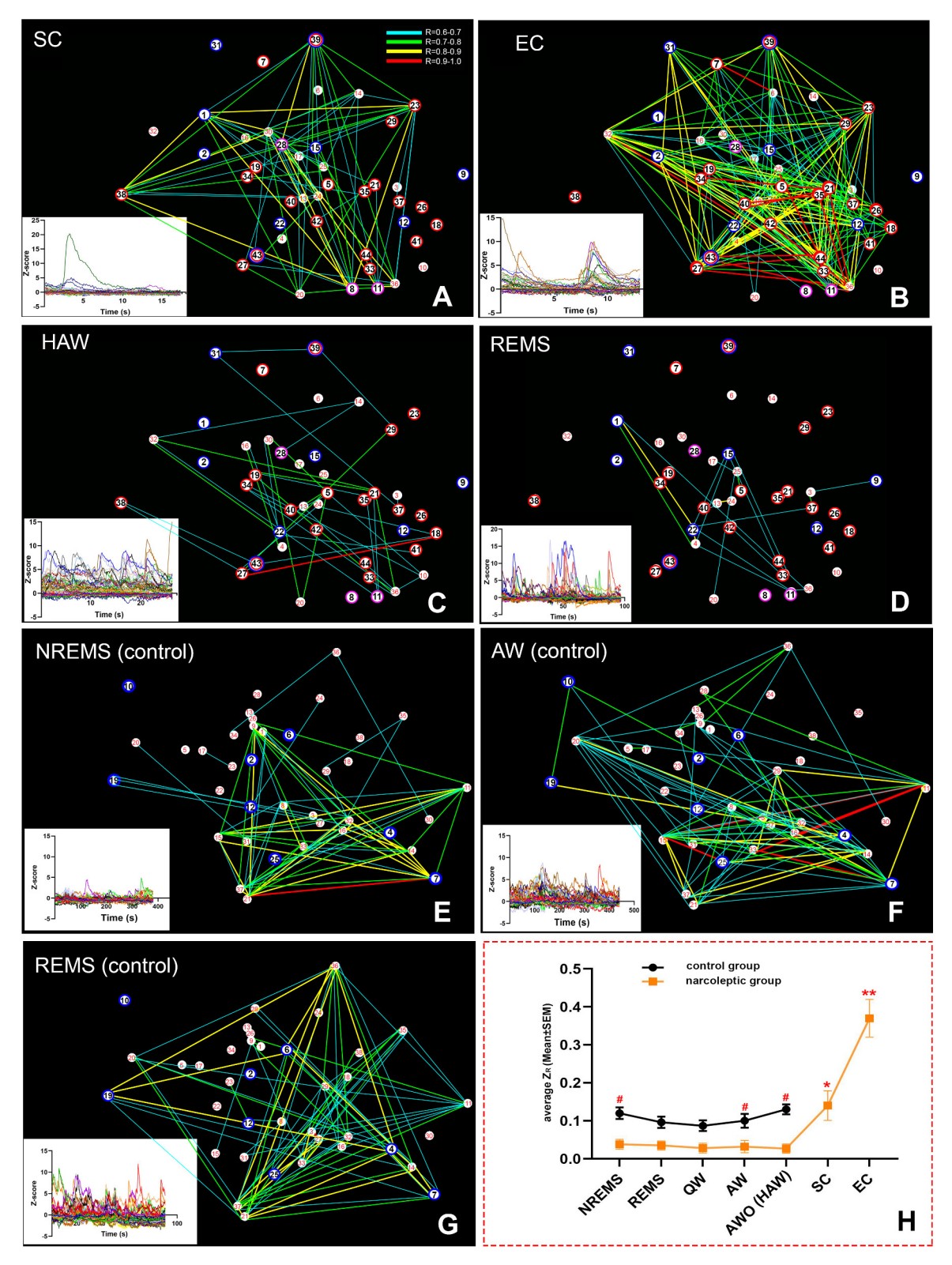

**Figure 5.** Spatial connectivity maps and graphs. Colored lines represent a strong positive correlation R value between 0.6 (blue) to 1.0 (red). (A–D): correlation maps of an SC (A), EC (B), HAW (C) and REMS (D) bout from narcoleptic mouse LR02082418. (E–G): correlation maps of an NREMS (E), AW (before odor exposure, (F) and REMS (G) bout from control mouse LR04101618. Ca$^{2+}$ signal intensity (ΔF/F Z-score) plots of corresponding brain states were placed on the bottom left of each panel. There were discernible dissociations between intensity (Z-score) and correlations R values. For instance,

*Figure 5 continued on next page*

*Figure 5 continued*

HAW bout in (C) had the highest Z-scores while had the least correlation lines, leaving many cells unconnected. (H): the summary of the average $Z_R$ (corrected R) trends of both groups during various brain states. Overall, narcoleptic mice had significantly lower $Z_R$ than the control group mice during undisturbed sleep/wake states. However, this low $Z_R$ status was abruptly reversed during cataplexy. $Z_R$ during SC was back to wild type level while $Z_R$ during EC went far higher than the wild type level. #: $p<0.05$ as compared to narcoleptic mice. *: $p<0.01$ as compared to other states of narcoleptic mice. **: $p<0.001$ as compared to other states of both groups. EC-ON, SC-ON and REMS-ON cells were circled red, purple and blue, respectively.

DOI: https://doi.org/10.7554/eLife.48311.017

The following source data is available for figure 5:

**Source data 1.** source data for correlation maps and graph.

DOI: https://doi.org/10.7554/eLife.48311.018

orexin neurons would be more like a complicated modulator to maintain the hemostasis on both activity intensity and correlations in the amygdala. The absence of orexin caused not only hyperactivity but also a failure to stabilize the rhythms of neuronal activities. As a result of this failure, CeA GABAergic neurons stayed desynchronized until challenged by emotion-induced cataplexy, in which CeA GABAergic neurons presented a hyper synchronization. The involving mechanism resulting in this hyper synchronization is still unclear. More evidence is needed to confirm the direct causal relationship between hyper synchronization and cataplexy. If hyper synchronization does facilitate cataplexy, instead of using strong inhibition, which oppresses the normal functions of the amygdala, desynchronizing might be a better way to alleviate emotion-induced cataplexy.

## Cataplexy and REMS

Cataplexy has long been considered as an inappropriate REMS intrusion into waking because of the similarities on EEG features and 'muscle atonia' symptom, which is usually observed during REMS (***Roth et al., 1969***). However, this hypothesis has never been experimentally verified. Narcolepsy patients and narcoleptic animals still have an ultradian rhythm of REM sleep. Stimulating cholinoceptor in the basal forebrain of narcoleptic dogs induced cataplexy attack but did not block the cyclicity of burst of rapid eye movement, implying separate mechanisms underlying cataplexy and REMS (***Nishino et al., 2000b***). Furthermore, the close linking among emotions, REMS, and cataplexy remains mostly unclear. In the present study, we demonstrated that some amygdala GABAergic neurons involved in spontaneous cataplexy were also active in NREMS, but none were also active in REMS. Though many amygdala GABA neurons were actively involved in REMS, these RMES-ON neurons were not one of the major components of cataplexy-ON neurons during emotion-induced cataplexy. Instead, those 'unscored' neurons that were not previously activated in sleep became the major component of cataplexy-ON neurons with a highly activated and synchronized way. It is implying that these neurons might use brain circuitry, distinct from the one for REMS generating and regulating, to trigger muscle atonia during cataplexy.

Although the potential abnormalities of Amygdala GABAergic neurons before and during cataplexy were identified in this study, their effect on

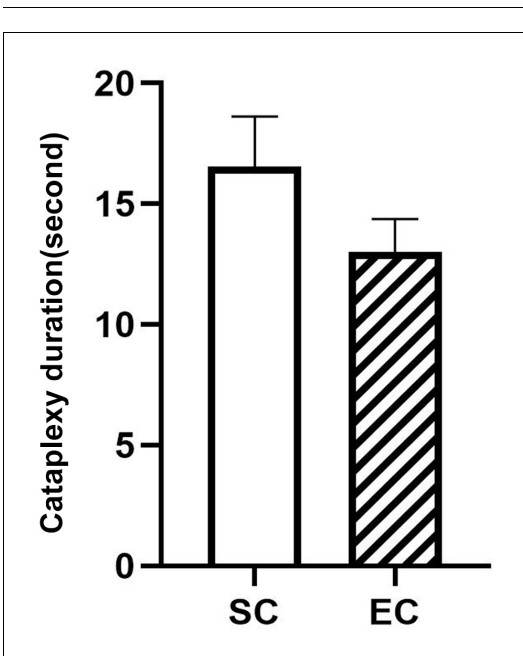

**Figure 6.** Durations of 10 spontaneous (SC) and 14 emotion-induced cataplexy (EC). The duration difference between SC and EC are insignificant ($F_{(1,22)}=3.02$, $p=0.09$).

DOI: https://doi.org/10.7554/eLife.48311.019

The following source data is available for figure 6:

**Source data 1.** Source data for duration graph.

DOI: https://doi.org/10.7554/eLife.48311.020

cataplexy duration is still unclear. We found that the difference in cataplexy duration between SC (n = 10) and EC (n = 14) has no statistical significance, though the SC duration seems longer than EC (*Figure 6*). Further studies are needed to understand the amygdala role in cataplexy maintaining.

## Technical considerations

The deep brain calcium imaging tool provides useful neural network information, but it does not come without any limitations. The weight of the miniature camera hindered the locomotion of small animals like mice, causing relatively fewer cataplexy attacks in our study when compared to other studies using group-housed mice with wheel running apparatus. The calcium sensor GCaMP6 itself could impose impairment on neurons by affecting L-type calcium channels of neurons (*Yang et al., 2018*). Improvement on light-weight camera, wireless recording, and developing harmless calcium sensors like GaMP-X (*Yang et al., 2018*) will make the deep-brain calcium imaging a much useful tool for behavioral studies.

## Conclusions

Cataplexy-ON neurons in amygdala were first identified in the dog (*Gulyani et al., 2002*). Now we have discovered them in narcoleptic mice and determined that the majority of these neurons are maximally activated right before and during emotion-induced cataplexy. Both the hyperactivity and hyper synchronization of CeA GABAergic neurons might contribute to emotion-induced cataplexy.

# Materials and methods

**Key resources table**

| Reagent type (species) or resource | Designation | Source or reference | Identifiers | Additional information |
|---|---|---|---|---|
| Genetic reagent (*M. musculus*) | VGAT-Cre | The Jackson laboratory | RRID: MGI:5141285 | PMID:21745644 |
| Genetic reagent (*M. musculus*) | Orexin KO | PMID:10481909 | RRID: IMSR_KOMP: VG11744-1.1-Vlcg | Dr. Masashi Yanagisawa (University of Texas) |
| Recombinant DNA reagent | AAV.Syn.Flex.G CaMP6s.WPRE.SV40 | http://www.addgene.org/100845/ | RRID: Addgene_100845 | PMID:23868258 Originally purchased from university of Pennsylvania gene therapy core |
| Antibody | anti-VGAT (Mouse monoclonal) | Synaptic System | Cat# 131011, RRID:AB_887872 | PMID:17444497 (1:500) |
| Antibody | anti-Orexin (Goat polyclonal) | Santa Cruz Biotechnology | Cat#: SC-8070, RRID:AB_653610 | PMID:16100511 (1:5000) |
| Software, algorithm | SPSS | IBM | RRID:SCR_002865 | |
| Software, algorithm | Matlab | MathWorks | RRID:SCR_001622 | |

## Animals and surgery

All manipulations done to the mice followed the policies established in the National Institutes of Health Guide for the Care and Use of Laboratory Animals and were approved by the Medical University of South Carolina Institutional Animal Care and Use Committee (protocol # IACUC-2019–00723).

To specifically target GABAergic neurons in narcoleptic mice, orexin KO mice (*Hcrt* [-/-]) mice (derived from founders donated by Dr. Yanagisawa, Southwestern Medical Center, Dallas, TX) were crossed with VGAT-Cre Knock-In mice (*Slc32a1-ires-Cre*[+/+], www.jax.org, stock #016962, Bar Harbor, ME). Offspring with the confirmed genotype *Slc32a1-ires-Cre*[+/-]/*Hcrt* [-/-] were used as the narcoleptic group (n = 10, both sexes, 6–10 months of age) while littermates with the confirmed genotype *Slc32a1-ires-Cre*[+/-]/*Hcrt* [±] were selected as the control group (n = 8, both sexes, 6–10 months of age). Genotype validation on mice tail snips was done off-site by Transnetyx (Cordova, TN). The temperature in the mice housing/recording room was always maintained at 23–25℃ under a 12 hr

light/dark cycle (lights on at 6:00 A.M.). Mice were given *ad libitum* access to regular laboratory food and water.

Under deep anesthesia (isoflurane 1.0–2.0%) and using a stereotaxic frame (Kopf, Tujunga, CA), AAV vectors with Cre inducible expression of GCaMP6 slow (AAV5-CAG-DIO-GCaMP6s, Titer: 3.48 $\times$ $10^{13}$ genomic copies/ml; University of Pennsylvania Preclinical Vector Core) were microinjected unilaterally into the CeA at the following coordinates: 1.11 mm posterior to Bregma, 2.95 mm lateral to the sagittal suture, and 4.30 mm ventral to the brain surface (*Liu et al., 2011*). Viral vectors were delivered in a volume of 500 nl using a 10.0 µL Hamilton syringe coupled to a 33-gauge stainless steel injector (Plastics One, Roanoke, VA). Injections were done gradually over 15 min. After microinjection, the injector needle was left in place for 15 min and then withdrawn slowly. At this time, and following the same injection track, a miniature Gradient Refractory INdex lens (GRIN, O.D. diameter: 0.6 mm, length: 7.3 mm; Inscopix Inc, Palo Alto, CA) was driven into the brain just above the CeA and cemented to the skull. Then, and as described elsewhere (*Liu et al., 2011*), four small screw-type electrodes and a pair of plate-type electrodes (Plastics One, CA) were implanted onto the mouse skull and nuchal muscles for recording the electroencephalogram (EEG) and electromyogram (EMG) activity respectively (*Figure 1*). Ten days after GRIN lens placement, mice were deeply anesthetized again (1.0–2.0% isofluorane). A baseplate was attached to a single photon miniaturized fluorescence microscope/CCD camera (nVoke from Inscopix, Inc, CA). The miniaturized microscope, along with the baseplate, were carefully placed atop the GRIN lens. The distance between the miniaturized microscope and the GRIN lens top was precisely adjusted until fluorescent neurons came into focus. At this focal point, the baseplate was secured around the GRIN lens cuff with dental cement, and then the microscope was detached. To protect the GRIN lens from debris and scratches, a cap was secured onto the baseplate. One week later mice were habituated to the recording experiment setting for three consecutive days before the sleep and $Ca^{2+}$ recording started.

## Sleep recording and identification of sleep states or cataplexy

After being amplified and filtered (0.3–100 Hz for EEG; 100–1 K Hz for EMG, MP150 system; Biopac Systems Inc, CA), the EEG/EMG signals were acquired and synchronized to the imaging of the $Ca^{2+}$ transients. In parallel, a night-vision camera was used to record mouse behavior. Streaming video of the mouse behavior was also synchronized with imaging of the $Ca^{2+}$ transients (Neuroscience Studio acquisition software, Doric Lenses Inc, QC, Canada). A MATLAB (Mathworks Inc, Natick, MA, USA) script was used to plot the spectrogram of the EEG activity (1 s window size and 0.5 s overlap).

EEG/EMG data (as CSV files) along with synchronized behavior video files were then transferred into SleepSign software (KISSEI Comtec Ltd., Nagano, Japan) and scored in 4 s epochs as wakefulness, non-rapid eye movement sleep (NREMS), REMS, and cataplexy. Wakefulness was identified by the presence of desynchronized EEG coupled with high amplitude EMG activity and further divided into quiet wakefulness (QW) or active wakefulness (AW) depending on whether the mouse displayed behaviors such as walking, rearing, grooming, eating, drinking, digging (AW), or was just standing still but awake (QW). Thus, during QW, the mouse kept an immobile posture often interspersed with movements of the head (e.g., bobbing) but did not exhibit any further purposeful movement. QW often occurred in between, in anticipation of, or following sleep states and lasted from a few seconds to several minutes. NREMS was scored when the EEG showed high-amplitude/low-frequency waves (delta waves) together with a lower EMG activity relative to waking. REMS was identified by the presence of regular EEG theta activity coupled with very low EMG activity.

To be qualified as a cataplexy attack, an episode had to meet the following criteria: 1) An abrupt episode of nuchal atonia lasting at least 8 s. 2) Immobility during the episode. 3) Theta activity dominant EEG during the episode. 4) At least 40 s of wakefulness preceding the episode (discrete cataplexy) or the first episode when several cataplexy episodes occur sequentially. The above criteria were slightly modified from the International Working Group on Rodent Models of Narcolepsy (*Scammell et al., 2009*).

## Miniature microscopy $Ca^{2+}$ transients imaging

At 10:00 AM, the mouse was gently restrained (swaddled in Terrycloth), while the miniature fluorescent microscope/camera was attached to its baseplate. At the same time, a lightweight cable was

plugged to record the EEG/EMG signals. The tethered mouse was then returned to the home cage and allowed to adapt for 6 hr for three consecutive days. On the fourth day (recording day), the same adaptation routine was followed, but at 4:00 PM, $Ca^{2+}$ transients-derived fluorescence began to be imaged by the nVoke miniaturized microscope/CCD camera (Inscopix, CA) and collected by its acquisition software. $Ca^{2+}$ associated fluorescence was continuously generated by a blue LED (power: 0.2 mW) and imaged at 10 frames per second (fps). To synchronize the timestamps of $Ca^{2+}$ imaging with the EEG/EMG, a TTL signal was sent from the nVoke interface console into the Doric console. Mouse was exposed to the predator odor coyote urine between 7:00 PM - 8:00 PM. Briefly, 1.0 ml coyote urine (www.predatorpee.com, Bangor, ME), stored in a 5 ml plastic vial filled with cotton, was placed in the home cage at 7:00 PM for 1 hr (*Liu et al., 2016*).

## Analysis of $Ca^{2+}$ transients imaging data

The person analyzing data was blind to the mice's genetic background, which has not been decoded until all analyses finished. $Ca^{2+}$ transient data were processed off-line by the Inscopix data processing software (version 1.1.2). Briefly, raw movies were first pre-processed to correct for defective pixilation, row noise and dropped frames. Preprocessed movies were then corrected for motion artifacts to generate the steadiest $Ca^{2+}$ fluorescent signals. The motion-corrected movies were subsequently mean filtered. To normalize the $Ca^{2+}$ signals, a single frame average projection of the filtered movie was generated. The average frame was used as the background fluorescence (F0) to calculate the instantaneous normalized $Ca^{2+}$ fluorescent signals ($\Delta F/F$) according to the formula; $(\Delta F/F)_i = F_i - F0/F0$ where i represents each movie frame. The normalized movie or '$\Delta F/F$ movie' was then used for semiautomatic extraction of $Ca^{2+}$ fluorescent signals associated with individual cell based on the principal and independent component analysis (PCA-ICA). Regions of interest (ROIs) identified by PCA-ICA were visually selected as candidate cells based on $\Delta F/F$ and image (cell-morphology). To be chosen as *bona-fide* neurons, $Ca^{2+}$ traces had to fulfill the canonic $Ca^{2+}$ spike waveform featuring fast-rising onset followed by slower decaying signal. $Ca^{2+}$ trace ($\Delta F/F$) of each ROI (cell) was further standardized as Z-score using the mean and standard deviation (SD) of each cell's $\Delta F/F$ (Z-score = $(\Delta F/F - Mean)/SD$). Since the lowest Z-score values were observed during QW, and no cell reached its maximal activity during QW, we used the average Z-score of QW ($Z_{QW}$) as the baseline. If a cell has an average Z-score during a specific state equal to or greater than ($Z_{QW}$ +1), it is scored as an 'ON' cell in that state. After cells were completely scored, we ran the ANOVA with Bonferroni post-hoc test to confirm that the average Z-score of these 'ON' cells were statistically higher than that of those 'non-ON' cells. In the meantime, $\Delta F/F$ Z-scores expressed as the percent of its maximal Z-score value across the whole recordings were plotted on a heat map (Sigmaplot software, Systat Software Inc, San Jose, CA). We then used the percentage threshold, which was defined as 80% of the maximum Z-score value during the whole recording period, to detect the prominent neuronal peak events, and set 2 s as the minimum interval between two adjacent peaks. The cumulative probabilities of peak frequency were compared between AW and AWO in the narcoleptic group with Kolmogorov-Smirnov test.

## Synchronicity and cross-correlations analysis of amygdala GABAergic neurons

First, the Z-score data were processed with Pearson correlation analysis (two tails) with Prism eight software (GraphPad Software, San Diego, CA), to obtain the pairwise correlation coefficient R between every two cells. Coefficient R is the index for correlation ranging from −1 to 1 (0 represents no correlation, −1.0 represents a negative correlation. 1.0 represents a perfect correlation). Next, we transferred the R to $Z_R$ based on the Fisher's Z transformation formula ($Z_R = 0.5[\ln(1 + R) − \ln(1 R)]$) so that the $Z_R$ became normally distributed. The average $Z_R$ of all positive correlations (R > 0) in each cell was calculated for statistical comparisons among groups and states. Finally, to show the correlation results intuitively, we generated the 2D spatial maps of all recorded cells and linked every two cells with color-coded lines representing the R values (strong positive correlations with R between 0.6–1.0).

## Histology

At the end of the study, the mice were anesthetized with isoflurane (5%) and perfused transcardially with 0.9% saline (5–10 ml) followed by 10% buffered formalin in 0.1M PBS (50 ml). Mice brains were harvested and cross-sectioned at 40 μm thickness (four sets) on a compresstome (Precisionary Instruments, Greenville, NC). To visualize the GRIN lens track and the location of the GCaMP6s transgene expression, coronal sections were scanned on a Leica fluorescent microscope. Mice that had a main GCaMP6s expression area outside of the amygdala were excluded from further analysis. To verify that GCaMP6 was expressed exclusively in GABAergic neurons, VGAT immunostaining was performed on one set of brain sections. Briefly, sections were incubated at room temperature for 24 hr with mouse anti-VGAT monoclonal antibody (1:500 dilutions, Synaptic System, Germany), followed by 1 hr incubation with Alexa fluor-568 donkey anti-mouse IgG (1:500, Invitrogen, CA). GCaMP6 and VGAT positive cells were counted on digitized images using MCID image analysis software (St. Catharines, ON, Canada). To confirm the correctness of the genotyping, orexin immunostaining was made on a separate set of brain sections. Briefly, sections were incubated at room temperature for 24 hr with goat anti-orexin polyclonal antibody (1:5000 dilutions, Santa Cruz Biotechnology, CA), followed by 1 hr incubation with biotinylated donkey anti-goat IgG (1:500, Millipore, Burlington, MA) secondary antibody and finally labeled using ABC–DAB–nickel staining (Vector Laboratories, Burlingame, CA).

## Statistical analysis

Generalized linear mixed model (GLMM) analysis (SPSS, version 25) with unconstructed covariance and Sequential Bonferroni post-hoc tests were used to compare the means of Z-score among each sleep/wake state inside or between two animal groups. One-way or two-way ANOVA and Bonferroni post-hoc test was used to compare the $Z_R$ values, cataplexy duration, peak amplitude and frequency. Statistical significance was evaluated at the $p < 0.05$ (two-tailed) level (*Kirk, 1968*).

## Acknowledgements

We thank Dr. Priyattam Shiromani for the advice, guidance, and technical support that made it possible to complete the study. We thank him for providing the miniscope and GRIN lenses for this study. Supported by NIH grants 1R01NS096151, 1R21NS101469.

## Additional information

### Funding

| Funder | Grant reference number | Author |
|---|---|---|
| National Institute of Neurological Disorders and Stroke | 1R01NS096151 | Meng Liu |
| National Institute of Neurological Disorders and Stroke | 1R21NS101469 | Meng Liu |

The funders had no role in study design, data collection and interpretation, or the decision to submit the work for publication.

### Author contributions

Ying Sun, Data curation, Software, Validation, Investigation, Writing—original draft; Carlos Blanco-Centurion, Conceptualization, Methodology, software, Writing—review and editing; Emmaline Bendell, Software, Investigation, Methodology; Aurelio Vidal-Ortiz, Siwei Luo, Methodology, Writing—review and editing; Meng Liu, Conceptualization, Formal analysis, Supervision, Investigation, Methodology, Writing—review and editing

### Author ORCIDs

Meng Liu (iD) https://orcid.org/0000-0003-1394-5014

## Ethics

Animal experimentation: All manipulations done to the mice followed the policies established in the National Institutes of Health Guide for the Care and Use of Laboratory Animals and were approved by the Medical University of South Carolina Institutional Animal Care and Use Committee (protocol # IACUC-2019-00723). All surgery was performed under isoflurane inhalation, and every effort was made to minimize suffering.

## Decision letter and Author response

Decision letter https://doi.org/10.7554/eLife.48311.023
Author response https://doi.org/10.7554/eLife.48311.024

## Additional files

### Supplementary files

• Transparent reporting form
DOI: https://doi.org/10.7554/eLife.48311.021

### Data availability

All data generated or analysed during this study are included in the manuscript and supporting files.

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
