## [Decision Letter]

Thank you for submitting your article "Amygdala GABAergic neuron activity dynamic during cataplexy of narcolepsy" for consideration by *eLife*. Your article has been reviewed by three peer reviewers, including Yang Dan as the Reviewing Editor and Reviewer #1, and the evaluation has been overseen by Huda Zoghbi as the Senior Editor.

The reviewers have discussed the reviews with one another and the Reviewing Editor has drafted this decision to help you prepare a revised submission.

Summary:

Sun et al. measured calcium activity of CeA GABAergic neurons across different vigilance states including spontaneous and emotionally-activated cataplexy using narcoleptic mice and micro-endoscopic calcium imaging. They measured activity of CeA GABAergic neurons in spontaneous cataplexy and predator odor-induced cataplexy. The author concluded that abnormal activation and synchronization of CeA GABAergic neurons trigger emotion-induced cataplexy. The study addresses an important question. Although the observations are very interesting, some claims are not well supported by the results. In addition, the messages of the individual figures as well as in the texts are not clearly described. The manuscript also contains many typo and grammatical errors that should be corrected before publication.

Essential revisions:

- The authors should perform more thorough statistical analysis and show more data to demonstrate that activation of CeA neurons is really correlated with cataplexy attacks. E.g. are SC- and EC-ON cells consistently activated during cataplexy attacks? SC-ON neurons might be activated just by chance during spontaneous cataplexy attacks and EC-ON neurons might not be specifically correlated with the attack, but rather activated by the predator odor (which might be still present in the recording chamber during the attack or the activity of the neurons might have a slow off dynamics and consequently overlap with the attack).

- The difference between spontaneous cataplexy and predator odor-induced cataplexy (e.g., duration and EEG spectrum) is not well described. What is definition of two different cataplexy states? How did the author distinguish spontaneously cataplexy which may occur during coyote urine application? To compare previous reports, this definition is important. For example, it is well known that palatable food such as chocolate induced cataplexy. The authors also need to provide a clear definition of cataplexy attacks (spontaneous vs. odor-induced), a more thorough discussion of the mouse model, and more detailed analyses of HAW behaviors.

- The manuscript lacks detailed information to evaluate the importance of findings. For example, there are no confirmation and quantitative data that GCaMP6 was exclusively expressed in the GABAergic neurons. Please provide GCaMP6s expression in vGAT-expressing neurons and expression ratio in the CeA and other brain regions include BLA.

- one needs to be very careful with the apparent correlation between neurons seen in single-photon imaging, because of neuropil contamination. This concern is particularly strong when many neurons in the field of view are activated. Much more careful analyses are needed to support the claim.

- The claim that abnormal activation and synchronization of CeA GABAergic neurons trigger emotion-induced cataplexy is not supported by any data. The authors need to tone down their conclusion, and if they wish to claim that abnormal activation and synchronization of CeA GABAergic neurons trigger emotion-induced cataplexy, they need to inhibit CeA and test its effect.

---

## [Author Response]

Essential revisions:- The authors should perform more thorough statistical analysis and show more data to demonstrate that activation of CeA neurons is really correlated with cataplexy attacks. E.g. are SC- and EC-ON cells consistently activated during cataplexy attacks? SC-ON neurons might be activated just by chance during spontaneous cataplexy attacks and EC-ON neurons might not be specifically correlated with the attack, but rather activated by the predator odor (which might be still present in the recording chamber during the attack or the activity of the neurons might have a slow off dynamics and consequently overlap with the attack).

We understand the reviewers’ concern that some neurons may just be activated during cataplexy, or any other states, by pure coincidence. We did observe some neurons showing strong activity in some episodes of a brain state (such as REM sleep) while staying silent during some other episodes of the same brain state. So we did not use an individual “activity peak” as the criteria to identify “ON cells.” Instead, we used significantly elevated average Ca^2+^ intensity of all episodes during each state, as the threshold, to reduce the bias from pure coincidence. For example, an SC-ON neuron was only scored when it had a significantly higher average activity level during all SC episode in this mouse.

As for the EC-ON neurons: We agree that the HAW-ON and EC-ON neurons were activated, indeed, by the predator odor, but it does not exclude the possibility that they were still specifically correlated to EC. It may follow the serial connection that the odor activates these neurons then these neurons subsequently trigger cataplexy. The current study is simply to describe the neuronal activity pattern during odor exposure and cataplexy. We do agree that further studies are needed to verify the direct causal effect between EC-ON neurons and emotion-induced cataplexy.

To further confirm the abnormal hyperactivity of CeA GABAergic neurons, we updated Figure 4E-J to demonstrate the significant increases on Z-score, Peak amplitude, and frequency after odor exposure in the narcoleptic mice. The original Z-scores were also plotted in Figure 4—figure supplement 1, to show the increased activity level.

- The difference between spontaneous cataplexy and predator odor-induced cataplexy (e.g., duration and EEG spectrum) is not well described. What is definition of two different cataplexy states? How did the author distinguish spontaneously cataplexy which may occur during coyote urine application? To compare previous reports, this definition is important. For example, it is well known that palatable food such as chocolate induced cataplexy. The authors also need to provide a clear definition of cataplexy attacks (spontaneous vs. odor-induced), a more thorough discussion of the mouse model, and more detailed analyses of HAW behaviors.

Exogenous emotion stimulus is sufficient to trigger cataplexy but not necessary for cataplexy. Cataplexy that happens without any exogenous emotion stimulus is called spontaneous cataplexy (SC), while those induced by palatable food or aversive odor are called emotion-induced cataplexy(EC). But, so far, there are no exclusive behavior or neurochemistry criteria for distinguishing between SC and EC. We, for the first time, used a hyperactive active waking (HAW) event as the criteria. If a cataplexy bout is led by a HAW then it is defined as an EC. Otherwise, it is defined as an SC. Most important, we found that EC and SC may have distinct neural substrate at the amygdala level. We have added more descriptions in the manuscript on how to define HAW and emotion-induced cataplexy with one-sample T-test. We are currently testing that if the palatable food or odor could also produce similar hyperactivity in the amygdala in the narcoleptic mice during cataplexy.

Since we only recorded for an hour after odor exposure, we did not see any SC during the odor exposure duration. But we believe it is possible to see an SC in the presence of the odor given the exposure duration is so long that animal becomes fully adapted to the odor.

In our previous study (2006) using coyote urine, we did not see significant differences in EEG spectrum and duration between control night and coyote urine night in the same orexin-KO mice. In this study, we calculated the average duration of 10 SC and 14 EC and found no significant difference (see Figure 6).

- The manuscript lacks detailed information to evaluate the importance of findings. For example, there are no confirmation and quantitative data that GCaMP6 was exclusively expressed in the GABAergic neurons. Please provide GCaMP6s expression in vGAT-expressing neurons and expression ratio in the CeA and other brain regions include BLA.

This VGAT-Cre mice model has been widely used in neuroscience research to target GABAergic neurons. Here we performed VGAT immunostaining with a cytoplasm VGAT antibody. New pictures showing VGAT/GCaMP6 double-labeled neurons were added in Figure 1. >95% of GCaMP6 expressing neurons were also containing VGAT immunoreactivities (see Figure 1—source data 1 for the counts).

- one needs to be very careful with the apparent correlation between neurons seen in single-photon imaging, because of neuropil contamination. This concern is particularly strong when many neurons in the field of view are activated. Much more careful analyses are needed to support the claim.

We agree with the reviewers that correlation may be affected by neuropil contamination, especially when there are so many neurons activated in the field of view. The Inscopix data analysis software contains a pre-processing component that removes out some of the noise from neuropil. After that, the spatial filter process also removes low-frequency noise. We recorded 27-44 cells in each field of review. This was an intermediate density allowing decent distance among most of the recorded cells, that helped keep the neuropil contamination minimal. We are currently working with collaborators to develop a customized tool for completely eliminating neuropil contaminations in our future studies.

- The claim that abnormal activation and synchronization of CeA GABAergic neurons trigger emotion-induced cataplexy is not supported by any data. The authors need to tone down their conclusion, and if they wish to claim that abnormal activation and synchronization of CeA GABAergic neurons trigger emotion-induced cataplexy, they need to inhibit CeA and test its effect

We reworded the Discussion conclusion part according to the reviewers’ suggestion.